# Engineering Rapalog-Inducible Genetic Switches Based on Split-T7 Polymerase to Regulate Oncolytic Virus-Driven Production of Tumour-Localized IL-12 for Anti-Cancer Immunotherapy

**DOI:** 10.3390/ph16050709

**Published:** 2023-05-07

**Authors:** Nikolas T. Martin, Mathieu J. F. Crupi, Zaid Taha, Joanna Poutou, Jack T. Whelan, Sydney Vallati, Julia Petryk, Ricardo Marius, Bradley Austin, Taha Azad, Mason Boulanger, Tamara Burgess, Ilson Sanders, Camille Victoor, Bryan C. Dickinson, Jean-Simon Diallo, Carolina S. Ilkow, John C. Bell

**Affiliations:** 1Centre for Innovative Cancer Research, Ottawa Hospital Research Institute, Ottawa, ON K1H 8L6, Canada; 2Department of Biochemistry, Microbiology and Immunology, University of Ottawa, Ottawa, ON K1H 8M5, Canada; 3Department of Chemistry, The University of Chicago, Chicago, IL 60637, USA

**Keywords:** Split-T7 RNA polymerase, interleukin-12, chlorotoxin, oncolytic vaccinia virus

## Abstract

The approval of different cytokines as anti-neoplastic agents has been challenged by dose-limiting toxicities. Although reducing dose levels affords improved tolerability, efficacy is precluded at these suboptimal doses. Strategies combining cytokines with oncolytic viruses have proven to elicit potent survival benefits in vivo, despite promoting rapid clearance of the oncolytic virus itself. Herein, we developed an inducible expression system based on a Split-T7 RNA polymerase for oncolytic poxviruses to regulate the spatial and temporal expression of a beneficial transgene. This expression system utilizes approved anti-neoplastic rapamycin analogues for transgene induction. This treatment regimen thus offers a triple anti-tumour effect through the oncolytic virus, the induced transgene, and the pharmacologic inducer itself. More specifically, we designed our therapeutic transgene by fusing a tumour-targeting chlorotoxin (CLTX) peptide to interleukin-12 (IL-12), and demonstrated that the constructs were functional and cancer-selective. We next encoded this construct into the oncolytic vaccinia virus strain Copenhagen (VV-*iIL-12mCLTX*), and were able to demonstrate significantly improved survival in multiple syngeneic murine tumour models through both localized and systemic virus administration, in combination with rapalogs. In summary, our findings demonstrate that rapalog-inducible genetic switches based on Split-T7 polymerase allow for regulation of the oncolytic virus-driven production of tumour-localized IL-12 for improved anti-cancer immunotherapy.

## 1. Introduction

Immunotherapies for cancer are undeniably the pharmaceutical technology of the future. This has been extensively demonstrated by the success of immune checkpoint inhibitors (reviewed in [1,2,3]), chimeric antigen receptor (CAR) T cells (reviewed in [4,5]), and oncolytic viruses (OVs, reviewed in [6,7]). Originally thought to only selectively infect and lyse tumour cells by exploiting aberrations in the antiviral response of these cells, OVs have proven to be much more than ‘just’ viruses that infect cancer cells. By eliciting immune responses within the tumour microenvironment (TME), these biotherapeutics can effectively turn an immunological ‘cold’ tumour into a ‘hot’ one, thereby reinvigorating the immune system of a cancer patient to fight the disease [6]. Today, OVs have little in common with their wildtype, usually pathogenic parental strains; they have been evolved or engineered into highly modifiable and safe delivery platforms that allow molecular biologists to delete unfavorable genes for attenuation [8,9,10]. With a growing understanding of both the TME and the underlying virology of OVs, we can now heavily modify the genome of OVs to alter their viral properties and allow for multiple transgene insertions. The range of therapeutic transgenes is continuously expanding, ranging from (modified) cytokines, antibodies, miRNAs, shRNAs, bispecific T cell engagers (TCEs), and prodrug-converting enzymes to whole virus genomes (reviewed in [11,12,13,14,15,16]). However, some transgenes can be cytotoxic or possess antiviral properties, posing a serious challenge in the production and manufacturing of recombinant OVs.

One way to circumvent these obstacles is the use of an inducible expression system. More specifically, the expression of a transgene can be regulated by the presence (or absence) of an inducer, usually a small molecule with a known safety profile. Several groups have shown the feasibility of this technology in the context of OV therapy [17,18,19]. Thus far, most controlled expression systems have been derived from the tetracycline resistance mechanism found in Gram-negative bacteria [20] and use either antibiotic tetracycline as an inducer or one if its derivatives, such as doxycycline (reviewed in [21,22]). Expression systems derived from this mechanism are well-characterized and often used in molecular sciences due to their tight gene regulation and the use of an FDA-approved drug. Another notable inducible expression system that has been used to drive gene expression from an OV is the GeneSwitch system, which utilizes a recombinant transactivator, partially based on the human progesterone receptor, and the drug Mifepristone as an inducer [18,23].

A potential downside of the more widely used tetracycline-controlled expression system lies in its inducer: Antibiotics can alter the microbiome, and can therefore heavily change the outcome of immunotherapies, with growing evidence of them being rendered ineffective [24,25,26,27]. Another important downside can be the antiviral effect of doxycycline on some OV vectors [28]. We therefore wanted to develop a novel inducible expression system that is independent of host cellular factors and that utilizes a different class of drug, ideally one that can act synergistically with the oncolytic virus [29] or provide anti-cancer effects on its own. Based on a biosensor involving a Split-T7 RNA Polymerase (RNAP) that dimerizes, and therefore becomes activated, in the presence of rapamycin [30], we developed this system into an inducible expression system to be used for an oncolytic vaccinia virus, and in combination with the FDA-approved and anti-cancer rapamycin-analogs (rapalogs) [31,32,33] Everolimus or Temsirolimus as inducers. This system offers several advantages, including the use of anti-cancer drugs as inducers and the use of T7-RNA polymerase, which offers high fidelity [34] and long transcripts in the vaccinia virus [35,36,37]. Furthermore, the system recognizes a small, 19 bp promoter, and does not rely on any host factors [38]. This Split-T7 RNAP comprises two components: the smaller N-terminal part of the RNAP (179 amino acid, named 29-1), which is fused at its C-terminus to the FKBP-rapamycin binding domain (FRB) by a serine–glycine linker, and the remaining C-terminal part of the RNAP (180-883 aa), which is N-terminally fused to the FK506 binding protein (FKBP) by a serine–glycine linker as well. Both parts of the RNAP have been modified by phage-assisted continuous evolution to reduce interaction in the absence of an inducer. The proteins FRB and FKBP are a well-described small-molecule-induced dimerization system [30].

Here, we generated several recombinant poxviruses and showed robust upregulation of different transgenes in the presence of inducer drugs both in vitro and in vivo. One such transgene is interleukin-12 (IL-12), a pro-inflammatory type 1 cytokine that forms a heterodimeric complex composed of p40 and p35 subunits [39]. The secretion of IL-12 by antigen-presenting cells generally shifts the immunosuppressive tumour microenvironment (TME) towards a milieu that supports both the recruitment and effector functions of CD8+ T and NK cells [40,41]. Of note, systemic IL-12 administration in cancer patients is poorly tolerated [42], and its delivery via engineered T cells is associated with severe toxicity [40]. We therefore predicted that OV-driven and the rapalog-inducible production of IL-12 would contribute to safe and potent anti-tumour immune responses that can be regulated in mice bearing aggressive breast, colorectal, or ovarian tumours. To further limit the systemic release and improve the localization of secreted IL-12 to the TME, we targeted the cytokine with a chlorotoxin (CLTX) 36 aa peptide [43], which serves as a TME-targeting domain for CAR T cell therapy by broadly recognizing many tumour types and has minimal cross-reactivity with non-malignant cells [44]. CLTX was isolated from the venom of the death stalker scorpion [45], and primarily utilizes the matrix metalloprotease-2 (MMP-2) as its receptor [46], a surface molecule found on a high percentage of glioblastoma cells and a wide variety of other tumours [45]. While doing so, CLTX has shown minimal cross-reactivity with non-malignant cells in the brain or elsewhere [45]. Ultimately, we showed potent efficacy of our novel oncolytic virus in preclinical mouse models, in the presence of rapalog inducer, which boosts the expression of CLTX-targeted IL-12.

## 2. Results

### 2.1. Split-T7 RNAP Can Be Encoded into the Genome of Poxviruses and Induces Transgene Expression When Supplemented with Rapamycin

As a proof-of-concept, we generated a recombinant modified vaccinia Ankara (MVA) virus that encodes the components of the Split-T7 RNAP biosensor (FRB-C-term and N-term-FKBP) together with a reporter gene (FLuc-P2A-EGFP) under the control of the 19 bp T7 promoter (Figure 1a). As the kinetics of this novel system were unknown, MVA was chosen for the initial experiments due to its inability to complete a viral life cycle in mammalian cells [47], allowing for transgene expression from the virus genome before cancer cell lysis; this can occur more slowly with MVA in contrast to other poxviruses [48], thus allowing for a longer time to induce transgene expression. When used to infect BHK-21 cells, MVA can spread via cell-to-cell contact, leading to the formation of viral foci, which can readily be observed via fluorescence microscopy (Figure 1b). The components of the Split-T7 RNAP were cloned into the MVA181R gene locus (homologous to VV Copenhagen B14R). Each part of the split polymerase was placed under the control of a strong, constitutively active vaccinia promoter. The FRB-C-terminal part was placed under the control of the artificial early–late (pE/L) promoter [49] and was encoded in the 5′ to 3′ direction. Downstream of this coding sequence (cds), the gene for N-term-FKBP was placed under the control of the synthetic late–early optimized promoter (pLEO) [50] and was encoded in the 3′ to 5′ direction. Between the genes coding for both components of the Split-T7 RNAP, we placed the cds for mCherry under the control of the p7.5 promoter [49] as a selection marker for virus purification, flanked by loxP sites for subsequent removal via Cre-mediated recombination. The reporter gene, driven by the T7-promoter, was encoded within the MVA086R (J2R) gene locus, disrupting the gene for thymidine kinase (TK) and causing functional knock out. Knocking out TK is a well-described, safety-enhancing modification of oncolytic VVs [8,51], and the TK gene locus is a good harbor for the integration of large transgenes, as shown by our own previous research. As reporters, we chose Firefly luciferase (FLuc) and EGFP, with the expression of the latter tied to FLuc expression via a self-cleaving P2A peptide [52]. As a selection marker for purification of the recombinant virus, we chose TagBFP under the control of the H5R promoter [53,54]. The cds for TagBFP was flanked by FRT sites for subsequent removal via FLP-mediated recombination (Figure 1a). Using these reporter genes, we were able to show an induction of EGFP expression in BHK-21 cells infected with MVA-*iFLucEGFP* when supplemented with rapamycin at concentrations ranging from 10 nM to 1 µM (Figure 1b). In brief, BHK-1 cells were infected with MVA-*iFLucEGFP* at a multiplicity of infection (MOI) of 0.1 plaque forming units (pfu)/cell (MOI = 0.1 pfu/cell) and supplemented with rapamycin at the indicated concentrations 48 h post-infection (hpi). Viral foci can be readily identified at 72 hpi through the expression of mCherry, the constitutively expressed selection marker for the B14R gene locus. Encouraged by these results, we moved the expression system to the oncolytic strain VV Copenhagen [55].

### 2.2. The Anti-Cancer Drugs Everolimus and Temsirolimus Can Be Used as Inducers of Transgene Expression

A major advantage of this novel expression system is the ability to utilize a small-molecule inducer that not only does not interfere with the line of treatment for a cancer patient, but in fact, can synergize by using the already clinically relevant anti-cancer drugs everolimus and temsirolimus. Using the same expression system as that described in Section 2.1, this time encoded into the genome of the oncolytic VV strain Copenhagen (VV-*iFLucEGFP*), we infected BHK-21 cells at MOI = 0.1 pfu/cell. We then treated the cells at 24 hpi with either DMSO, the vehicle as a negative control, rapamycin as a positive control, everolimus, or temsirolimus at concentrations from 1 nM to 50 nM (DMSO was added in equal volumes). The increase in EGFP fluorescence with increasing amounts of inducer was readily observable (Figure 2a,b). The induction of expression using 10 nM of either rapamycin, everolimus, or temsirolimus led to an approximate five-fold increase in EGFP fluorescence (Figure 2b), while the level of fluorescence measured for the constitutively expressed mCherry remained unchanged (Figure 2c). Comparable levels of induction could be observed when the inducer was supplemented at the time of infection (Appendix A).

### 2.3. VV-iFlucEGFP Can Infect Cancer Cell Lines, and Transgene Expression Can Be Induced In Vitro and In Vivo

After demonstrating the functionality of the Split-T7 RNAP-inducible expression system using BHK-21 cells, we evaluated its effect on the human colorectal adenocarcinoma cell line HT-29, which can be readily used for xenograft models, as we have previously described [56]. First, we showed the infectivity and induction of transgene expression in vitro in a similar fashion to that described in Section 2.2. HT-29 cells were infected at MOI = 0.1 pfu/cell with VV-*iFLucEGFP*, and transgene expression was induced at 24 hpi with rapamycin, everolimus, and temsirolimus at concentrations from 1 nM to 50 nM (DMSO was added in equal volumes as a negative control). An increase in EGFP fluorescence was detected via fluorescence microscopy, in contrast to the constitutively expressed mCherry, which served as a marker for virus infection (Figure 3a, Appendix A). Overall, rapamycin, everolimus, and temsirolimus can be used to induce transgene expression in colon carcinoma cells infected with VV-*iFLucEGFP* to a similar extent to that observed previously using non-human BHK-21 cells (Figure 2). The observable decrease in fluorescence of both mCherry and EGFP when treated with 25 nM to 100 nM (Figure 3a, Appendix A) was caused by an increase in cell death, as observed via visual inspection using light microscopy. Next, we assessed the induction of transgene expression in vivo. We therefore seeded 2 × 10^6^ HT-29 cells subcutaneously in the flanks of athymic nude mice and treated tumours measuring approximately 171.5 mm^3^ via intratumoural (IT) injection with VV-iFLucEGFP at 1 × 10^7^ pfu on days 4 and 3, prior to baseline IVIS imaging (Figure 3b). After baseline imaging, the animals were treated once with an IP injection of 10 mg/kg rapamycin, everolimus, or temsirolimus, respectively. Luminescence within the tumours was then imaged 1, 2, and 5 days post-baseline (Figure 3b,c). Overall, there was an approximate 10-fold increase in luminescence in animals treated with rapamycin or the rapalogs compared to the vehicle control, which lasted for approximately 5 days post-treatment (Figure 3c,d). Of note, tumour volumes in animals treated with everolimus and temsirolimus started to noticeably decrease at the end of the experiment.

### 2.4. Design and Validation of Tumour-Targeted IL-12 for Expression of VV for Cancer Therapy

We designed a tumour-targeted IL-12 construct by fusing a CLTX peptide to an IL-12 p35 subunit of the p70 full-length cytokine, in order to facilitate bystander killing through proinflammatory immune-mediated cell death (Figure 4a). Before encoding this transgene into a viral-inducible system, we validated that two versions of our chimeric protein (Appendix A) could be expressed upon virus infection via immunoblotting (Appendix A). These include IL-12mCLTX and IL-12hCLTX, which herein refer to CLTX fusion to either murine IL-12 or human IL-12 (which serves as a non-functional control for in vivo studies), respectively. We also verified that IL-12-CLTX versions were functionally active using an HEKBlue™ IL-12 reporter assay (Appendix A). To determine whether the CLTX peptide enhanced the retention of IL-12 in tumours, we generated a vaccinia virus (VV-IL-12m) that encodes an untethered version as a control (Figure 4b). Both VV-IL-12mCLTX and VV-IL-12m underwent HiBiT-tag fusion to the cytokine transgene to allow for detection in serum and tumours. We therefore injected mice bearing MC38 tumours with 1 × 10^7^ pfu of either virus and collected the tumours and serum after 24 h. We detected a significant increase in IL-12 retention in the tumours ex vivo when the IL-12 was fused to CLTX (Figure 4c). No differences were observed in the luminescence of the serum. These data suggest that CLTX recognizes and binds to MMP-2 in MC38 tumours, which express high levels of MMP-2 [57]. After validating the optimal design, we generated inducible versions in the vaccinia viruses (VV-*iIL-12mCLTX* and VV-*iIL-12hCLTX*) via the Split-T7 strategy, as described in Section 2.1. Next, we infected U-2 OS cells with VV-*iIL-12mCLTX* and VV-*iIL-12hCLTX* viruses and co-treated them with everolimus, temsirolimus, or rapamycin at 50 nM. Cell lysates were collected at 24 hpi and immunoblots were probed for IL-12 (Figure 4d). These data show that the viruses led to a significant increase in the production of IL-12mCLTX or IL-12hCLTX in the presence of rapalogs compared to cells treated with the DMSO vehicle control. To determine whether the therapeutic transgenes were also being secreted, we collected the supernatants of infected cells and performed a split-nanoluciferase assay. Briefly, the CLTX peptide in our constructs was tagged with HiBiT technology, which allows for transgene detection via LgBiT/NanoGlo-mediated nanoluciferase reconstitution, as we have previously described for TCEs [56]. We detected a significant increase in nanoluciferase signal from the supernatants of cells treated with rapalogs or rapamycin compared to the DMSO-treated controls (Figure 4e). To ensure that the IL-12 levels were not increased by changes in virus replication in the presence of the drugs, we titered the viral output at 24 hpi (Figure 4f). As predicted, we did not detect any differences in VV-*iIL-12mCLTX* replication in U-2 OS cells in the presence or absence of rapamycin or rapalogs compared to the DMSO-treated controls. These data therefore confirm that the increase in IL-12 production is due to Split-T7 regulation, similarly to the increase in FLuc and EGFP described in Section 2.3. Similar to previously tested colorectal and breast cancer cells [56], we demonstrated in vitro that ovarian cancer cell lines are susceptible to VV infection but resistant to VV-mediated cytotoxicity, as detected via a resazurin assay (Appendix A). Among these human ovarian cancer cell lines (SKOV-3, OVCAR-4, OVCAR-5, OVCAR-8, and A2780) and ovarian cancer patient-derived ascites-fluid cell lines (AF2028 and AF2068), SKOV-3 cells were the most resistant and showed the smallest reduction in cell viability at 48 hpi. Interestingly, we showed via immunoblotting that SKOV-3 cells express the most abundant levels of MMP-2, similarly to murine ID8 Tp53−/− Pten −/− Fluc-expressing cells (Appendix A). These findings suggest that these ovarian cancer cell lines may allow for enhanced recognition and binding to CLTX, as previously demonstrated using glioblastoma and other cancer cell lines. Indeed, we detected IL-12mCLTX binding to the surface of SKOV-3 cells using a binding assay, as previously described [56], and HiBiT detection (Appendix A). In contrast, we did not detect binding of murine IL-12 in the absence of the CLTX peptide or cytokine-HiBiT control. Lastly, we showed that different cancer cell lines, including SKOV-3 cells (Appendix A), can be infected in vitro by VV-*iFLucEGFP*, and allow for transgene induction by rapamycin and rapalogs at 50 nM, as shown in Section 2.3. Our data suggest that ovarian, colorectal and breast cancer models are suitable for exploring the in vivo anti-cancer effects of VV-*iIL-12mCLTX*.

### 2.5. VV with Induced IL-12 Can Prolong Survival of Mice Bearing Ovarian Tumours

Using immunocompetent C57BL/6J mice, we evaluated the anti-tumour effects of VV-*iIL-12mCLTX*, VV-*CTRL* (no cytokine), or a PBS control. Mice bearing intraperitoneal ovarian ID8 Tp53 −/− Pten −/− Fluc tumours were injected intraperitoneally with three doses of the viruses at 1 × 10^7^ pfu, or the PBS as control, on days 6, 7, and 8 (Figure 5a). Mice were co-injected with everolimus at 10 mg/kg or the vehicle control on days 7 and 8. Despite the aggressive onset of disease, as measured via luminescence (Appendix A), mice bearing ovarian tumours treated with the combination of VV-*iIL-12mCLTX* and everolimus showed significantly prolonged survival (Figure 5b). These findings suggest that our VV with inducible CLTX-IL-12 may show benefits in other aggressive cancer models.

### 2.6. VV with Induced IL-12 Can Synergize with Immune Checkpoint Inhibition and Cure Mice with Peritoneal Carcinomatosis

Using immunocompetent C57BL/6J mice, we determined the anti-tumour effects of VV-*iIL-12mCLTX*, VV-*iIL-12hCLTX* (control virus with non-functional cytokine), and the PBS control. Mice bearing intraperitoneal colorectal MC38 tumours were injected intraperitoneally with four doses of the viruses at 1 × 10^8^ pfu, or the PBS control, on days 3, 4, 5, and 6 (Figure 6a). The mice were co-injected with everolimus at 10 mg/kg or the vehicle control on days 4, 5, and 6. Lastly, the mice were also treated with the αPD-1 or IgG control on days 4 6 and 12 14. In this model of peritoneal carcinomatosis, which we have previously characterized [56], tumours grow upon the gut, and the mice are euthanized prior to day 25 due to abdominal distension and respiratory distress. Despite the aggressive onset of disease, mice bearing MC38 tumours treated with the combination of VV-*iIL-12mCLTX* and everolimus showed significantly prolonged survival, and 20% of mice were cured compared to control groups (Figure 6b). In addition, survival was further prolonged with αPD-1 co-treatment compared to the IgG controls (Figure 6b). These findings suggest that our VV with inducible CLTX-IL-12 may synergize with immune checkpoint inhibition to generate long-lasting immune responses.

### 2.7. Intravenous Delivery of VV with Induced IL-12 Demonstrates Efficacy in Reducing Breast Cancer Metastases in Lungs

We studied the anti-tumour effects of our combination strategy in a breast cancer metastasis model of the lungs (Figure 7a). Following intravenous injection of 4T1.2 cells in BALB/c mice, we administered two doses of the viruses at 1 × 10^7^ pfu, or the PBS control, on days 1 and 2. We also intraperitoneally co-administered two doses of everolimus or the vehicle control at 10 mg/kg. The lungs were harvested on day 10 for staining (Figure 7b). In our breast cancer metastasis model, we showed that the combination of VV-*iIL-12mCLTX* (relevant cytokine) and everolimus significantly decreased the number of metastatic nodules in the lungs of BALB/c mice. Monotherapies did not significantly reduce metastases, except for VV-*iIL-12mCLTX* (relevant cytokine), suggesting that the basal or leaky expression of IL-12 provided some benefit. Of note, the in vivo induction of IL-12 by everolimus allowed VV-*iIL-12mCLTX* treatment to outperform all other combinations with PBS, VV-*iIL-12hCLTX* (control virus with non-functional cytokine), or the vehicle controls (Figure 7c).

## 3. Discussion

The notion that oncolytic virus therapy works mostly by infecting, lysing, and thus, destroying tumour cells has been reworked, and we now understand that OV vectors act more as immune modulators, activating the patient’s immune system and supporting its role in the fight against cancer [11,58]. Using modern molecular biology, we can rationally design OVs, allowing them to fulfill defined tasks within the tumour microenvironment. The most common approach to designing novel oncolytics is the encoding of therapeutic transgenes into the virus genome to be expressed within the tumour, acting on the cancer cells, residing immune cells, and potentially the oncolytic itself [55]. The expression of therapeutic transgenes, however, is limited by the spread of the OV within the tumour and is usually controlled by constitutively active viral promoters. This can be a problem for cytotoxic or antiviral transgenes, as constitutive expression would hinder virus spread, limiting not only the application of such a virus, but also its manufacture. One solution to overcoming this is the use of inducible expression systems.

In this proof-of-concept study, we have adapted a previously described biosensor [30] to be used as an inducible expression system for the oncolytic vaccinia virus. We have shown that this system can significantly upregulate the expression of both a reporter transgene as well as a novel therapeutic transgene. Our inducible expression system is based on the dimerization, and thus, activation of a Split-T7 RNA polymerase upon induction with rapamycin. Once active, the T7 RNAP can then transcribe any transgenes placed under the control of the T7 promoter with high fidelity and speed, both characteristics of this bacteriophage-derived RNAP [34,38]. We also demonstrated that not only rapamycin, but also the approved anti-cancer drugs and rapalogs [31,32,33] everolimus and temsirolimus, can be used in comparable concentrations as inducers. The use of these drugs not only negates the side effects of a treatment with rapamycin—namely immune suppression due to inhibition of the response to IL-2 [59], thus blocking the activation and proliferation of T and B cells—but, in fact, synergizes with the anti-tumour activity of the OV and the encoded therapeutic transgene.

Using the herein developed reporter virus VV-*iFlucEGFP*, we were able to show that the observable induction of transgene expression works comparably in cancer cell lines in vitro, and that this can also be observed in a xenograft in vivo model bearing HT-29 colorectal tumours. Interestingly, the observable induction levels were low when compared to the original publication on this biosensor [30] or the more commonly used tetracycline-inducible expression systems [21,22]. However, there are several explanations for this difference and multiple ways to further reduce basal or leaky expression, and thus, to improve the signal-to-noise ratio. In the first publication describing this biosensor, induction levels of over 340× were only observed in *E. coli*, and the components were transformed on separate plasmids [30]. In our setting, the components were expressed from constitutively active, strong viral promoters and within the viral replication factories within the cytosol of the infected cell. This potentially led to high localized concentrations of the RNAP components and may explain the observable background expression. Strategies to improve this are to carefully select the right viral promoter for the expression of the RNAP and to incorporate additional C-terminal modifications to the RNAP, as described in the follow-up publication of the original authors [60].

Cytokine-based immunotherapy has disappointingly only led to the approval of IFN-α and IL-2 as single agents [42], and GM-CSF, encoded in oncolytic herpes virus (T-VEC) [61], for the treatment of very few cancer indications. Recently, multiple efforts have been made in the field to revisit IL-12, despite its severe toxicities in early clinical trials and its minimal clinical responses [42]. To overcome these limitations, studies have focused on improving: (1) IL-12 targeting, (2) IL-12 delivery, and (3) the regulation of IL-12 production. To localize IL-12 to the tumour microenvironment, targeted IL-12 constructs have been engineered using antibodies or peptides that bind to cell-surface antigens on cancer or stromal cells. For example, an immunocytokine composed of two IL-12 heterodimers fused to human IgG1 (NHS76) was generated to enable IL-12 to target DNA/histone complexes, which are often exposed in necrotic tumours (e.g., NCT01417546 and NCT02994953). Several different delivery approaches have been explored in phase I/II clinical trials, including: (1) the plasmid-based delivery of IL-12 using lipopolymers (GEN-1) [62] for ovarian cancer patients (NCT02480374 and NCT03393884); (2) the virus-based delivery of IL-12 using oncolytic adenoviruses for metastatic pancreatic cancer (NCT03281382) and prostate cancer (NCT03330197), or oncolytic herpes simplex virus for brain tumours (NCT02062827); (3) the cell-based delivery of IL-12 using CAR-T cells for advanced malignant solid tumours (NCT03932565); and (4) the mRNA-based delivery of IL-12 using lipid nanoparticles for advanced solid tumours (NCT03946800 and NCT03871348). To modulate the production levels of IL-12 over time, an inducible adenoviral vector was designed to produce human IL-12 in the presence of the inducer Veledimex for brain cancers (e.g., NCT03330197, NCT02555397, NCT03330197, NCT03679754, NCT04006119, NCT03636477, and NCT02026271). Of note, our strategy combines all three of the above-mentioned approaches. We fused a CLTX-targeting moiety to IL-12 to promote its accumulation in the tumour microenvironment following systemic administration. To further limit off-target toxicity and side-effects (e.g., cytokine storm) in cancer patients, we encoded CLTX-IL-12 in an oncolytic vaccinia virus to allow for enhanced delivery in the tumour, and we controlled the production of CLTX-IL-12 by inducer drugs to further improve anti-cancer efficacy.

In immunocompetent mice, we demonstrated that an oncolytic vaccinia virus encoding inducible targeted IL-12 can significantly improve the survival of mice bearing aggressive ovarian or colorectal cancers, while reducing breast cancer metastases to the lungs, in combination with the approved rapalogs everolimus and temsirolimus. Overall, we showed that this novel inducible expression system holds great potential for further improvement, and certainly paves the way for other applications. The high fidelity, speed, and transcript length of the T7 RNAP allows for the control of multiple transgenes encoded separately on the same viral genome or the expression of multiple therapeutic payloads on one single, long transcript, separated by protein cleavage sites such as Furin or by self-cleaving peptides such as the P2A peptide used as within the reporter gene in this study. Lastly, the use of this system allows for the combination of OV therapy with rapalogs, which in this setting, are not just co-administered, as in the case of tetracycline-inducible expression systems, but actively synergize with the treatment of the cancer.

## 4. Materials and Methods

### 4.1. Constructs

The plasmids encoding the originally described Split-T7 RNAP biosensor, p7-68 and p7-69, were kindly provided by Bryan C. Dickinson. The coding sequences for the two components (29-1-FRB and FBKP-T7-C-term) were codon-optimized for human expression using GenSmart™ Codon Optimization and ordered in the pcDNA3.1+ expression vector (Genscript, Piscataway, NJ, USA). IL-12mCLTX and IL-12hCLTX were designed and also ordered in the pcDNA3.1+ expression vector (Genscript). Through restriction enzyme digestion, versions of the IL-12 constructs without CLTX were also generated.

### 4.2. Cell Culture

BHK-21 [C-13], U-2 OS, HeLa, DF-1 [CRL-12203], HT-29, SKOV-3, OVCAR-4, OVCAR-5, OVCAR-8, and A2780 cells (ATCC; Manassas, VA, USA) were cultured in either Roswell Park Memorial Institute/RPMI 1640 Medium (Gibco; Waltham, MA, USA) or Dulbecco’s Modified Eagle Medium/DMEM (GE Healthcare Life Sciences; Issaquah, WA, USA), supplemented with 10% fetal bovine serum/FBS (Gibco, ThermoFisher Scientific, Waltham, MA, USA). AF2028 and AF2068 ovarian cancer patient-derived ascites-fluid cell lines were gifted by Dr. Barbara Vanderhyden (OHRI, Ottawa, ON, Canada). ID8 Tp53−/− Pten −/− FLuc and 4T1.2 were gifted by Dr. Jean-Simon Diallo (OHRI, Ottawa, ON, Canada). MC38 cells were gifted by Dr. Guy Ungerechts (OHRI, Ottawa, ON, Canada). Cells were maintained in a humidified atmosphere at 37 °C in 5% CO_2_. Cells remained mycoplasma-free and were routinely tested via PCR (e-Myco™ VALiD Detection Kit, 25239, LiliF Diagnostics; Sungnam-si, Gyeonggi-do, Republic of Korea).

### 4.3. Rapamycin and Rapalog Preparation

For in vitro experiments, rapamycin and the rapalogs everolimus and temsirolimus were dissolved in DMSO, aliquoted, and stored at −20 °C. For in vivo experiments, drugs were dissolved in DMSO as described earlier, and diluted in 5% Tween-80 and 5% PEG-400 for no longer than 30 min prior to use.

### 4.4. T7-Promoter Driven Reporter and Transgenes

T7-RNA recognizes a 19 bp promoter sequence and generates an RNA transcript of everything downstream of this promoter sequence. Transcription is terminated by a 48 bp T7 termination signal. For initial in vitro testing, a reporter plasmid was generated (pRep-EGFP) that contained the coding sequence for EGFP under the control of the T7 promoter and flanked downstream by a T7 termination signal. This construct was ordered as a Gblock from IDT and cloned into a pUC19 vector using Gibson assembly. For generation of the recombinant OV, a recombination cassette was designed (pTK-Rep-EGFP) that contained the already-described T7 promoter-driven EGFP, flanked downstream by a T7 terminator, and a cds for TagBFP under control of the pH5R promoter as selectable fluorescent marker. Both coding sequences were flanked by 500 bp homologous regions for recombination into the J2R (thymidine kinase) locus of vaccinia. This construct was ordered, cloned into the pUC57 backbone (Genscript, Piscataway, NJ, USA), and linearized for recombination into the virus genome. Additionally, this plasmid was used to generate pTK-Rep-Fluc-P2A-EGFP, on which the cds of EGFP was replaced with the cds for Firefly luciferase coupled to EGFP by the self-cleaving P2A peptide, allowing for the quantification of induction levels via luminescence, as well as the convenient detection of protein expression via fluorescence microscopy. The IL-12 constructs were also ordered, cloned into the pUC57 backbone (Genscript), and linearized for recombination into the virus genome.

### 4.5. Recombinant Oncolytic Vaccinia Virus

For recombination into the VV Copenhagen genome, N- and C-terminal components were encoded under the VV promoters pEL and pLEO, respectively. Both coding sequences were oriented in a tail-to-tail orientation to minimize the chance of restoring a full-length T7 RNA polymerase via recombination and were also separated by the cds for the fluorescent selection marker mCherry, itself under control of the p7.5 promoter. All cds were flanked by 500 bp homologues regions for recombination into the B14R gene locus (see Figure 1a). This sequence was also ordered from Genscript. The recombinant virus was generated by transfecting the linearized recombination cassette into U-2 OS cells infected with the parental VV at MOI = 0.05, with subsequent plaque purification utilizing the mCherry fluorescent marker. Once a pure virus population was obtained, the reporter (or the inducible therapeutic transgene) was introduced into the J2R locus using the same method, selecting for TagBFP-mCherry double-positive plaques.

### 4.6. Oncolytic Virus Production and Titration

The parental VV Copenhagen strain and MVA (VR-1508) have been previously described [54]. For the production of oncolytic VV, HeLa cells were grown in DMEM supplemented with HEPES in 850 cm^2^ roller bottles, infected with the virus at an MOI of 0.05, and cultured until a cytopathic effect was observed (~72 h) at 37 °C. Viruses were collected and purified as previously described [56]. Viral titers were determined via titration using U-2 OS cells as previously demonstrated [56]. For the production of oncolytic MVA, chicken embryonic fibroblast DF-1 cells, which allow for MVA infection, were used instead of human HeLa and U-2 OS cells, and MVA was titered according to previously established methods [54].

### 4.7. Immunoblotting

Samples were prepared following protein quantification via a bicinchoninic acid assay (BCA kit, 23227; ThermoFisher Scientific, Waltham, MA, USA), and immunoblotting was performed with equal amounts of protein for whole cell lysates or supernatant samples as previously described [56]. Total protein was detected via ponceau staining (Ponceau S solution, P7170-1L; Sigma-Aldrich, St Louis, MO, USA). Human or murine IL-12 was detected with an IL-12 p35 antibody (1:1000, MAB1570; R&D Systems, Minneapolis, MN, USA). Human or murine MMP-2 was detected by an MMP-2 antibody (1:1000; Clone D2O4T, 87809S, Cell Signaling Technology, Danvers, MA, USA). Vaccinia virus presence was confirmed with a rabbit polyclonal antibody that detects vaccinia virus proteins (1:1000, LS-C103289; LSBio, Seattle, WA, USA). β-Actin (1:1000; 13E5; Cell Signaling Technology) was used as a loading control for immunoblots. After overnight incubation with primary antibodies, the immunoblots were probed with HRP-coupled anti-mouse or anti-rabbit antibodies (1:3000; Cell Signaling Technology) for 1 h and subsequently imaged on a Bio-Rad ChemiDoc.

### 4.8. In Vitro Assays

The metabolic activity of cancer cells was assessed using resazurin sodium salt (R12204; Thermo Fisher Scientific, Waltham, MA, USA), according to the manufacturer’s protocol. Treated and/or infected cells were administered 10% (*v*/*v*, final) resazurin in each well and incubated for 1–2 h, depending on the cell line. Fluorescence was measured at 590 nm upon excitation at 530 nm, using a BioTek Microplate Reader (BioTek, Winooski, VT, USA).

Virus infections were performed on cells in serum-free media at indicated MOIs. After inoculation for 1.5–2 h, the media were removed and replaced with RPMI or DMEM supplemented with 10% FBS and 1% (*v*/*v*) penicillin/streptomycin. After incubation at 37 °C in 5% CO_2_, cells were verified for EGFP expression at 24 or 48 hpi using an EVOS fluorescence microscope (Thermo Fisher Scientific) or Cellomics ArrayScan (Thermo Fisher Scientific).

IL-12 production/secretion was measured via enzyme-linked immunosorbent assay (mouse or human IL-12 p70 Quantikine ELISA Kit, M1270 or D1200; R&D Systems, Minneapolis, MN, USA) following the manufacturer’s instructions. For IL-12mCLTX binding assays, cells were placed on ice and treated with the concentrated/quantified IL-12 (HiBiT-tagged) for 1 h prior to washing away excess unbound IL-12 and measuring the luminescence to determine IL-12 attachment to the surface of cancer cells. For HiBiT detection, 1x passive lysis buffer (Luciferase Assay System Passive Lysis 5× Buffer, E1941; Promega, Fitchburg, WI, USA) was used to harvest/equalize whole cell lysates or supernatants before BCA. LgBiT and NanoGlo substrate were used as indicated in the manufacturer’s instructions (NanoGlo HiBiT Extracellular Detection System, N2421; Promega).

Human or murine IL-12 functionality was assessed using HEKBlue IL-12 reporter cells (hkb-il12; InvivoGen, San Diego, CA, USA), according to the manufacturer’s instructions. Briefly, HEKBlue IL-12 cells were grown in media with Normocin (ant-nr-1; InvivoGen), HEK-Blue selection buffer (hb-sel; InvivoGen), and heat-inactivated FBS. HEKBlue IL-12 cells were incubated with supernatants from transfected or infected (and filtered by 0.22 µm to remove VV) serum-starved cells containing IL-12 at indicated timepoints, and SEAP was detected using a BioTek Microplate Reader at 630 nm using Quanti-Blue solution (rep-qbs; InvivoGen). Recombinant human IL-12 (rcyc-hil12; InvivoGen) was used as a positive control.

### 4.9. Human Xenograft and Syngeneic Murine Model Studies

Prior to the initiation of treatment regimens, animal cohorts were randomized after tumour implantation. Cells for implantation were washed 2× with PBS, strained (0.45 µm filter), counted using ViCell, resuspended in PBS, and kept on ice until the administration of injections. HT-29 cells with Matrigel (2 × 10^6^ cells in 50 µL of PBS combined with 50 µL Matrigel, 356231; Corning, Canton, NY, USA) were injected subcutaneously in the right flanks of 8-week-old athymic nude mice (Charles River Laboratories, Wilmington, MA, USA). Tumour volumes were calculated using the following modified ellipsoidal formula: tumour volume = [(width^2^ × length)/2], where width is the smallest dimension. Intratumoural injections of the virus (2 doses at 1 × 10^7^ pfu in 100 µL of PBS), rapamycin/everolimus/temsirolimus (1 dose at 10 mg/kg in 100 µL of PEG formulation), or the vehicle control were administered at indicated timepoints. For imaging using an IVIS Spectrum (PerkinElmer, Waltham, MA, USA), mice were injected intraperitoneally with D-luciferin (molecular probes D-luciferin potassium salts, L2916; ThermoFisher Scientific, Waltham, MA, USA) at a dose of 10 mg/mL for 5 min prior to isoflurane anesthesia.

In 8-week-old immunocompetent C57BL/6J mice (The Jackson Laboratory; Bar Harbor, ME, USA), ID8 Tp53−/− Pten −/− FLuc murine ovarian cancer cells (5 × 10^6^ cells in 100 µL of PBS) were injected intraperitoneally into the abdomen. Treatments of intraperitoneal injections of the virus (3 doses at 1 × 10^7^ pfu in 100 µL of PBS) or of the PBS control, everolimus (2 doses 10 mg/kg) or the vehicle control were administered at indicated timepoints.

For the peritoneal carcinomatosis model, MC38 murine colorectal cancer cells (5 × 10^5^ cells in 100 µL of PBS) were injected intraperitoneally into the abdomens of C57BL/6J mice (the Jackson Laboratory). At indicated timepoints, the following treatments were administered: intraperitoneal injections of the virus (4 doses at 1 × 10^8^ pfu in 100 µL of PBS) or the PBS control; everolimus (3 doses 10 mg/kg in 100 µL of PEG formulation) or the vehicle control; and anti-PD-1 (InVivoMAb anti-mouse CD279 Clone RMP1-14, BE0146; Bio X Cell, Lebanon, NH, USA) or the IgG isotype control (InVivoMAb anti-trinitrophenol IgG2a Clone 2A3, BE0089; Bio X Cell) in 6 doses at 5 mg/kg in 100 µL of PBS. Tumour burden was monitored via IVIS imaging (PerkinElmer, Waltham, MA, USA) at indicated timepoints.

For the lung metastasis model, 4T1.2 murine breast cancer cells (5 × 10^5^ cells in 100 µL of PBS) were injected intravenously into the tail veins of BALB/c 8-week-old immunocompetent mice (Charles River Laboratories, Wilmington, MA, USA). Treatments of intravenous injections of the virus (2 × 10^7^ pfu in 100 µL of PBS) or the PBS control, and intraperitoneal injections of everolimus (10 mg/kg) or the vehicle control were administered at indicated timepoints before the lungs were harvested. At day 10, mice were euthanized, and their lungs were injected intratracheally with black India ink, excised, rinsed with water, and fixed in Fekete solution (100 mL formalin, 700 mL ethanol, 50 mL glacial acetic acid, ad 150 mL distilled water) before counting the metastatic nodules.

## 5. Statistical Analyses

Prism 9 (GraphPad) was used to perform all statistical analyses, and quantitative data are reported as means ± SEM, or as indicated in the figures. Statistical analyses were performed on raw data via one-way ANOVA to compare three conditions or more, two-way ANOVA with Tukey’s correction to compare groups influenced by two variables, multiple unpaired t tests, and the Kaplan–Meier method followed by a log-rank test for in vivo survival analyses. In the text or figure legends, the exact *p* values are provided, and differences between experimental groups were considered significant at *p* < 0.05.

## Figures and Tables

**Figure 1 pharmaceuticals-16-00709-f001:**
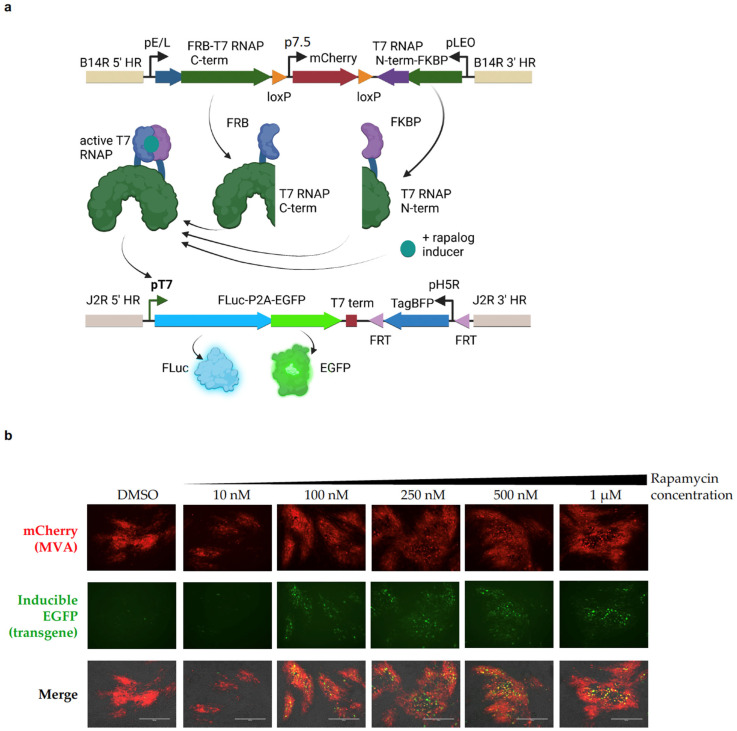
Generation of a Split-T7 RNAP-inducible expression system for poxviruses, including MVA. (**a**) Genomic layout of the Split-T7 RNAP-inducible expression system on the virus genome. Within the B14R gene locus (500 bp 5′ HR, beige), the cds for the FRB (dark blue)-T7 RNAP C-term (aa 180–883, dark green) is placed under the VV pE/L promoter and is encoded in sense. Downstream, flanked by loxP sites (orange), the cds for mCherry (dark red) is placed under control of the VV p7.5 promoter and acts as a constitutively expressed marker for selection of the recombinant virus. Downstream of this region, the cds for the T7 RNAP N-term (aa 1–179, dark green)-FKBP (purple) fusion protein is encoded in the antisense direction under control of the VV pLEO promoter. A second homologous region of 500 bp (B14R 3′ HR, beige) ensures site-directed integration into the virus genome. Both components of the inducible RNAP are constitutively expressed and dimerize (left) in the presence of a rapalog inducer (teal), forming the active T7 RNAP, which can now recognize the 19 bp T7 promoter (pT7, bold), and thus, transcribe the orf downstream of the promoter, in this case encoding both FLuc (light blue) and EGFP (green), which are coupled together by a self-cleaving P2A-peptide. This reporter gene is integrated into the J2R/TK gene locus by including two flanking homologous regions (5′ and 3′ HR, khaki) of 500 bp each. The selection of the recombinant virus is facilitated by encoding TagBFP under control of the VV pH5R promoter. This selection marker is flanked by FRT sites for subsequent removal. (**b**) The Split-T7 RNAP can transcribe a transgene under control of the T7 promoter in the presence of rapamycin. The viral foci of MVA-*iFLucEGFP* upon infection of BHK-21 cells can be readily observed via constitutive mCherry expression at 72 hpi (top row). Cells were supplemented with rapamycin at the indicated concentrations, ranging from 10 nM to 1 µM, or with DMSO (vehicle control) at 48 hpi. EGFP fluorescence was monitored at 72 hpi (middle row). An increase in EGFP fluorescence compared to the negative control (left column) and the 10 nM condition (second column) is observable under treatment with 100 nM rapamycin and higher. An overlay of transmitted light, mCherry, and EGFP fluorescence is given in the third row and shows the co-expression of mCherry and EGFP in the viral foci. Scale bar = 400 µm.

**Figure 2 pharmaceuticals-16-00709-f002:**
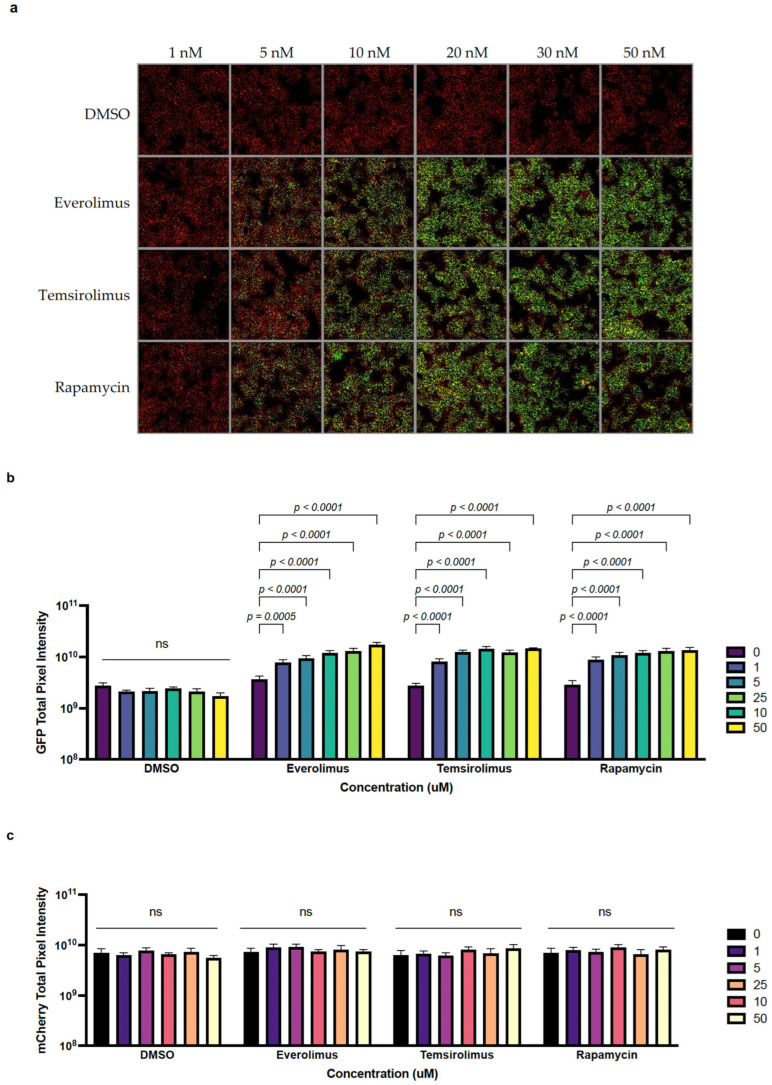
Split-T7 RNAP-inducible expression system for VV allows for in vitro EGFP and Fluc transgene induction by rapamycin and the FDA-approved rapalogs everolimus and temsirolimus. (**a**) The rapalogs everolimus and temsirolimus can induce expression at similar concentrations compared to rapamycin. BHK-21 cells were infected with VV-*iFLucEGFP* at MOI = 0.1, and were then treated with either equal volumes of DMSO (vehicle control, top row), everolimus, temsirolimus, or rapamycin at the indicated concentrations at 24 hpi. Constitutive mCherry expression and induced EGFP expression were monitored at 48 hpi using an ArrayScan. (**b**) An approximate 5-fold increase in EGFP fluorescence compared to the baseline was measured at an inducer concentration of 5 nM and higher, slightly increasing until it reached a plateau at 20 nM of inducer, regardless of the drug used. Shown are means ± SD. ns = not significant. Statistical analyses were performed using two-way ANOVA. (**c**) The rapalog inducers did not interfere with protein biosynthesis or viral spread, as indicated by an unchanged level of mCherry expression. Shown are means ± SD. ns = not significant. Statistical analyses were performed using two-way ANOVA.

**Figure 3 pharmaceuticals-16-00709-f003:**
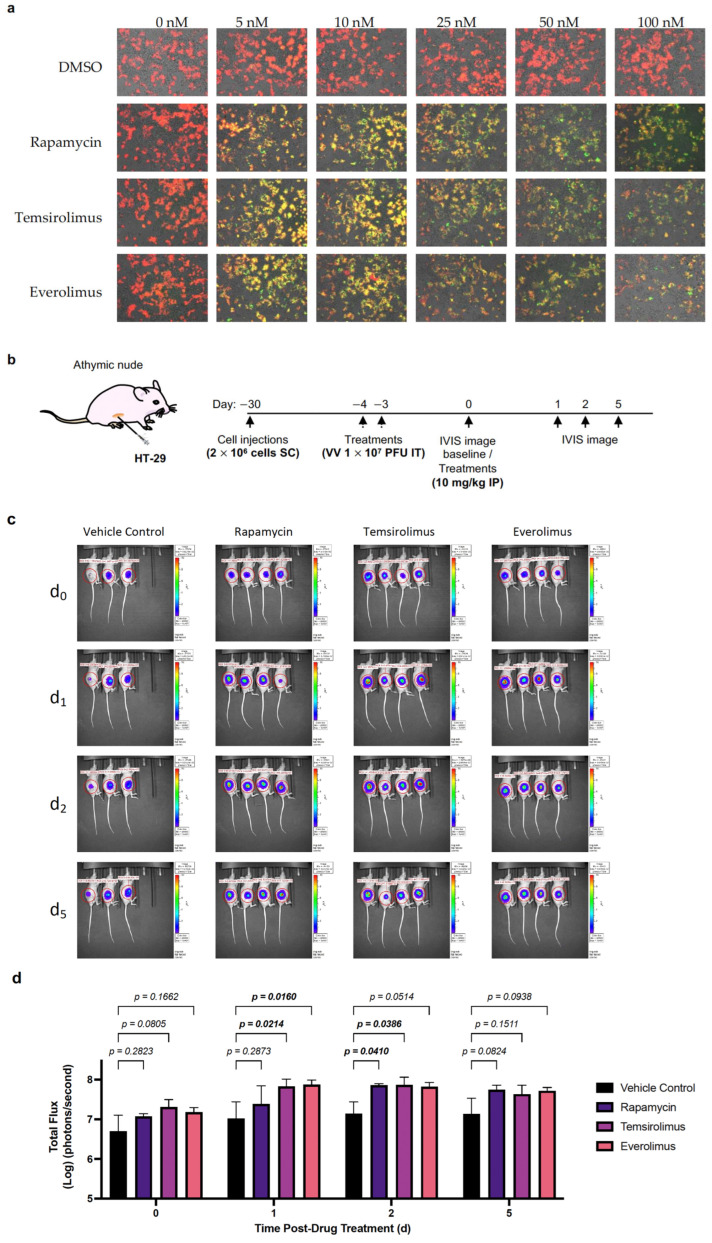
Split-T7 RNAP-inducible expression system for VV allowed for in vitro and in vivo transgene induction in colorectal tumour cells by rapamycin and the FDA-approved rapalogs everolimus and temsirolimus. (**a**) The human colon carcinoma cell line HT-29 was infectable with VV-*iFLucEGFP*, and transgene expression could be induced with rapalogs. HT-29 cells were infected with VV-*iFLucEGFP* at MOI = 0.1, and treated at 24 hpi with DMSO (vehicle control) in equal volumes, or a drug (either rapamycin, temsirolimus, or everolimus) at concentrations ranging from 0–100 nM. Viral infection was monitored at 48 hpi by measuring mCherry fluorescence. A significant increase in EGFP fluorescence over baseline is observable under treatment with as little as 5 nM of rapamycin, temsirolimus, and everolimus, respectively. A decrease in fluorescence is observed at higher concentrations (25 nM–100 nM) due to an overall decrease in cell viability. (**b**) Experimental timeline to show transgene inducibility in vivo in tumours colonized with VV-*iFLucEGFP.* Thirty days before baseline, athymic nude mice were injected in their flanks with 2 × 10^6^ HT-29 cells SC. Tumour growth was monitored over time, and once tumours reached approximately 171.5 mm^3^, they were injected twice with two doses of 1 × 10^7^ pfu of VV-*iFLucEGFP* 3 and 4 days prior to baseline imaging. On the day of baseline imaging (d0), animals were treated with an IP injection of 100 µL of either the PEG vehicle control or 10 mg/kg rapamycin, temsirolimus, or everolimus. (**c**) Luminescence was monitored via IVIS imaging 1, 2, and 5 days post-baseline imaging. (**d**) Overall, an approximate 10-fold increase in luminescence signal compared to the vehicle control is shown, lasting up to 5 days post-treatment. Statistical analyses were performed via two-way ANOVA.

**Figure 4 pharmaceuticals-16-00709-f004:**
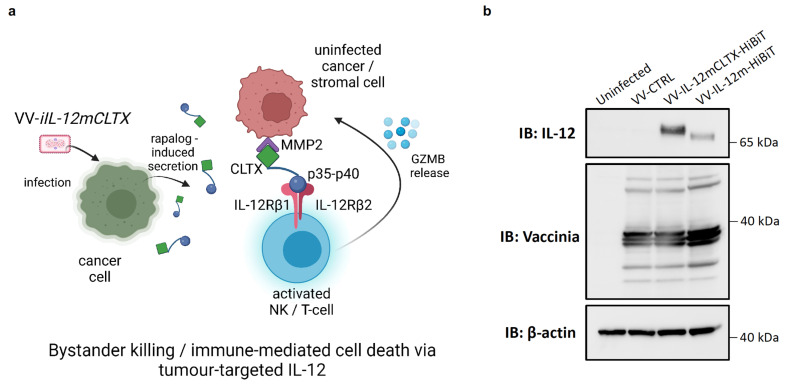
CLTX-targeted IL-12 can be encoded into VV, and its expression in cancer cells and its secretion can be induced via Split-T7 by rapamycin and rapalogs. (**a**) Schematic of a tumour-targeted IL-12 construct generated by fusing a CLTX peptide to the IL-12 p35 subunit (of the p70 full-length cytokine), in order to facilitate bystander killing through proinflammatory immune-mediated cell death. This construct was encoded into the vaccinia virus (VV-*iIL-12mCLTX*) via Split-T7 to allow for inducible production by FDA-approved rapalogs. (**b**) U-2 OS cells were treated with VV-IL-12mCLTX or VV-IL-12m viruses. Cell lysates were collected at 24 hpi. Immunoblots were probed for IL-12, VV, or β-Actin as a loading control. The untethered murine IL-12 (IL-12m) is smaller in molecular weight size compared to the version of IL-12 fused to CLTX (IL-12mCLTX), as expected. Vaccinia proteins are expressed at similar levels in infected cells, suggesting that the viruses do not replicate differently. (**c**) C57BL/6J mice bearing MC38 tumours were injected intratumourally with VV-IL-12mCLTX-HiBiT or VV-IL-12mCLTX-HiBiT at 1 × 10^7^ pfu, and tumours and serum were collected 24 hpi. Luminescence was quantified ex vivo through HiBiT-tag detection. ns = not significant. Statistical analyses were performed using multiple unpaired *t* tests. (**d**) U-2 OS cells were treated with VV-*iIL-12mCLTX* or VV-*iIL-12hCLTX* viruses and co-treated with everolimus, temsirolimus, or rapamycin at 50 nM. Cell lysates were collected at 24 hpi. Immunoblots were probed for IL-12, or for β-Actin as a loading control. Both viruses led to a significant increase in the production of IL-12mCLTX or IL-12hCLTX in the presence of rapalogs compared to cells treated with DMSO vehicle control. (**e**) Supernatants of infected cells were collected for a split-nanoluciferase assay. Incubation with LgBiT/NanoGlo allows for detection and quantification of the HiBiT tag fused to the CLTX peptide. We detected a significant increase in nanoluciferase signal (10-fold) from supernatants of cells treated with rapalogs or rapamycin compared to the DMSO-treated controls. Statistical analyses were performed via one-way ANOVA. (**f**) U-2 OS cells were infected with VV-*iIL-12mCLTX* and frozen at 24 hpi. Samples were freeze/thawed twice and titered using U-2 OS cells to measure viral output. VV-*iIL-12mCLTX* titers were not significantly (ns) altered in the presence or absence of rapamycin or rapalogs compared to the DMSO-treated controls. Statistical analyses were performed via one-way ANOVA.

**Figure 5 pharmaceuticals-16-00709-f005:**
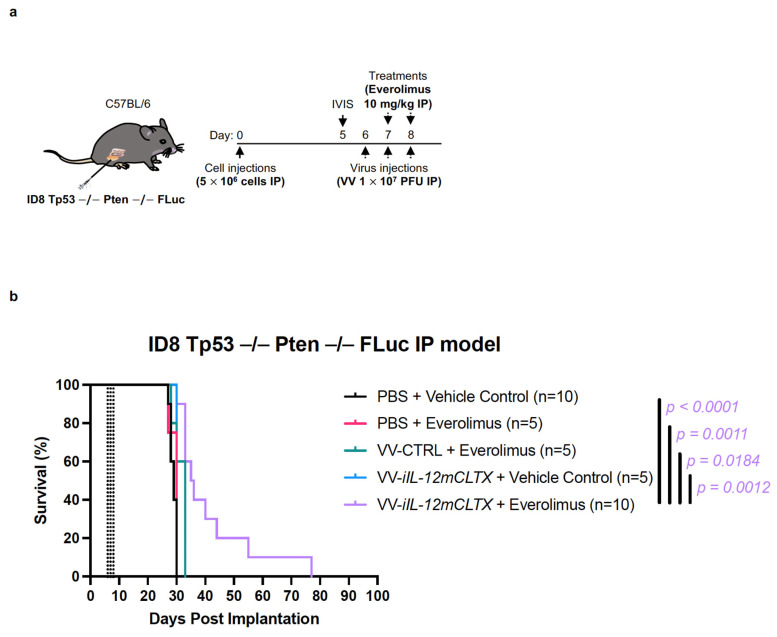
VV encoding an inducible IL-12mCLTX therapeutic transgene can prolong survival of immunocompetent mice bearing intraperitoneal ovarian tumours in combination with everolimus. (**a**) Anti-tumour effects of VV-*iIL-12mCLTX* (relevant cytokine), VV-*CTRL* (no cytokine), or PBS control were evaluated using immunocompetent C57BL/6J mice. Mice bearing intraperitoneal ovarian ID8 Tp53 −/− Pten − /− FLuc tumours were injected intraperitoneally with 3 doses of viruses at 1 × 10^7^ pfu, or PBS as control, on days 6, 7, and 8. Mice were co-injected with everolimus at 10 mg/kg or vehicle control on days 7 and 8. (**b**) Mice bearing ovarian tumours treated with the combination of VV-*iIL-12mCLTX* and everolimus showed significantly prolonged survival. Statistical analyses were performed using the Kaplan–Meier method, followed by a log-rank test.

**Figure 6 pharmaceuticals-16-00709-f006:**
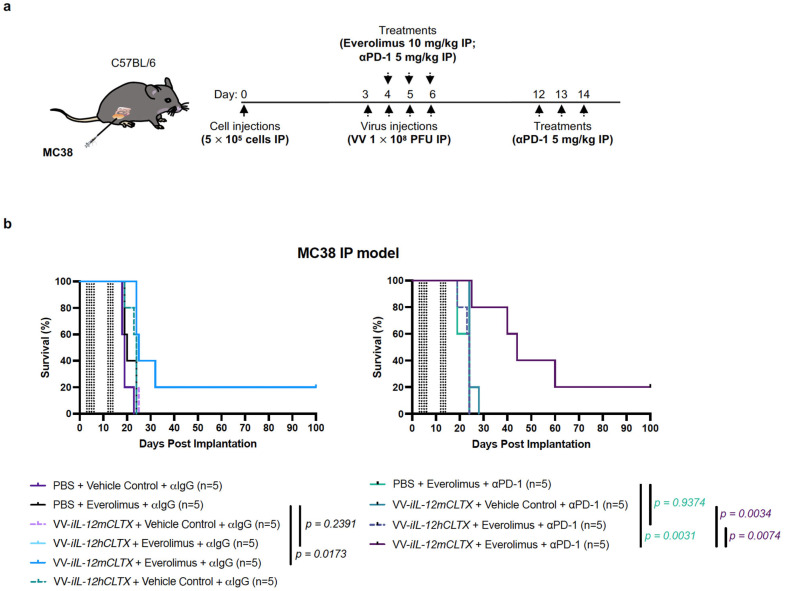
VV encoding an inducible IL-12mCLTX therapeutic transgene can prolong survival and cure immunocompetent mice bearing intraperitoneal colorectal tumours in combination with everolimus and immune checkpoint inhibitor. (**a**) Anti-tumour effects of VV-*iIL-12mCLTX* (relevant cytokine), VV-*iIL-12hCLTX* (control virus with non-functional cytokine), and PBS control were determined using immunocompetent C57BL/6J mice. Mice bearing intraperitoneal colorectal MC38 tumours were injected intraperitoneally with 4 doses of viruses at 1 × 10^8^ pfu, or PBS control, on days 3, 4, 5, and 6. Mice were co-injected with everolimus at 10 mg/kg or vehicle control on days 4, 5, and 6. Lastly, mice were also treated with αPD-1 or IgG control on days 4–6 and days 12–14. (**b**) Mice bearing MC38 tumours treated with the combination of VV-*iIL-12mCLTX* and everolimus showed significantly prolonged survival, and 20% of mice were cured compared to control groups. Survival was further prolonged with αPD-1 co-treatment compared to IgG controls. Statistical analyses were performed using the Kaplan–Meier method, followed by a log-rank test.

**Figure 7 pharmaceuticals-16-00709-f007:**
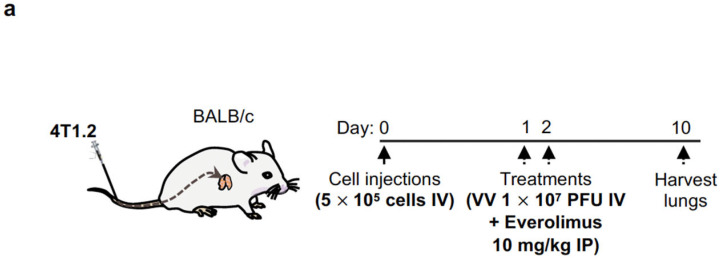
VV encoding an inducible IL-12mCLTX therapeutic transgene can reduce breast cancer metastases in combination with everolimus in lungs of immunocompetent mice. (**a**) The anti-tumour effects of our combination strategy were studied using an aggressive breast cancer metastasis model of the lungs. Following intravenous injection of 4T1.2 cells in BALB/c mice, we administered 2 doses of viruses at 1 × 10^7^ pfu, or the PBS control, on days 1 and 2. We also intraperitoneally co-administered two doses of everolimus or vehicle control at 10 mg/kg. (**b**) Lungs were harvested on day 10 for staining. In our breast cancer metastasis model, we showed that the combination of VV-*iIL-12mCLTX* (relevant cytokine) and everolimus significantly reduced the number of metastatic nodules in the lungs of BALB/c mice. Monotherapies did not significantly reduce metastases, except for VV-*iIL-12mCLTX* (relevant cytokine), suggesting that basal or leaky expression of IL-12 provided some benefit. (**c**) The in vivo induction of IL-12 by everolimus allowed VV-*iIL-12mCLTX* treatment to outperform all other combinations with PBS, VV-*iIL-12hCLTX* (control virus with non-functional cytokine), or vehicle controls. Statistical analyses were performed via one-way ANOVA.

## Data Availability

Data is contained within the article and Appendix A.

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
