# Peer review of "Engineering Rapalog-Inducible Genetic Switches Based on Split-T7 Polymerase to Regulate Oncolytic Virus-Driven Production of Tumour-Localized IL-12 for Anti-Cancer Immunotherapy"

_pharmaceuticals, 2023, doi:10.3390/ph16050709_

Round 1
Reviewer 1 Report
In this paper, the authors claim to have engineered a gene expression system that involves the use of FRB and FKBP fused with split-T7-RNA polymerase that can be activated by rapamycin or its analogues. They claim to use the system to control expression of an IL12 fusion protein from tumors to improve their clearance.
The data in figure 1 demonstrates the feasibility of the approach in a system that expresses GFP and luciferase. The addition of rapamycin seems to increase the number of GFP+ cells in the images presented. The increase in GFP+ cells appears to be concentration dependent. The data in figure 2 demonstrates the feasibility of the system using an oncolytic VV strain when it is activated using rapamycin analogues (Everolimus and Temsirolimus). The activation appears to be somewhat concentration dependent. Figure 3 evaluates the system in cancer cells in vitro and in vivo. Figure 4 evaluates the use of the system to express IL12 and IL12-CLTX fusion. They detect the expression of both human and mouse IL-12CLTX using western blotting and luminescence assays. The expression of IL12-CLTX is increased when cells are treated with rapamycin or its analogues. Figures 5-7 test the efficacy of the approach in clearing different types of tumors in immunocompetent mice. In figures 5 and 6, mice treated with the oncolytic virus and everolimus show improved survival compared to untreated controls and mice treated with oncolytic virus alone. The combination of this approach with anti-PD1 antibody further improves the efficacy. In figure 7, mice treated with this inducible IL12-CLTX expression system show fewer metastatic nodules. The data broadly supports the authors’ claims. Here are a few concerns
1. Figure 4B reports the specificity of the IL12-CLTX in targeting tumors. However, the comparison is made relative to serum concentrations. Is that the best comparison? The methods indicate that the samples were normalized for overall protein content using a BCA assay. But is it really possible to normalize protein content from serum with those from tissue? Or was the normalization performed differently? More clarification may help improve the paper further
2. T7 polymerase is from a bacteriophage. Do the authors expect a potential immune response against the polymerase? The peptides could be displayed by MHC complexes even the polymerase is not activated. Could this be a beneficial in improving the anti-tumor response? Is there a risk of off-target immune responses affecting other tissues? Has this polymerase been used in other OV therapies?
3. The system appears to be leaky based on the in vitro results. The control mice figure 3 also show luminescence. Given this data, it is a bit surprising that VV-IL12CLTX-vehicle controls did not show any improved survival in Figures 5 and 6. The authors have discussed the leakiness of the system. This data may provide some insight into its significance
4. In figures 1 and 2, the system expresses both Fluc and GFP. It is a bit challenging to appreciate the GFP expression in the images. Was luminescence measured? It would be nice to present the luminescence data to further validate the results.
Reviewer 2 Report
General comments to the paper entitled: Engineering rapalog-inducible genetic switches based on Split-T7 polymerase to regulate oncolytic virus-driven production of tumour-localized IL-12 for anti-cancer immunotherapy
The authors had an excellent job developing an inducible expression system. The transgene expression in the expression system was induced by anti-neoplastic rapamycin analogs. The system the authors built up included the oncolytic virus, the induced transgene, and the pharmacologic inducer, rapamycin. It was expected and proved by the experiments that the triple antitumor effect prolonged the survival of mice bearing ovarian and colorectal tumors and reduced breast cancer metastasis in the lungs.
This inducible expression system holds great potential for further improvement in immunotherapy.
I strongly suggest accepting and publish the paper without any modification.
Comments and suggestion to authors:
The experiments showed the efficacy of the immunotherapy by increasing the survival but did not stop the progression of diseases. It is known that the limiting factor improving the efficacy is the toxicity of the anti-neoplastic agent such as cytokines and chemotherapy. The new approach to develop anti-cancer drugs is the deuterium depletion. Deuterium depletion stimulates immune system, but deuterium depletion by inducing radicals triggers the apoptosis and leads to tumor necrosis. Combining deuterium depletion with the developed expression system may results an even better outcome.
Reviewer 3 Report
In the present manuscript, the authors designed a viral expression vector for potential application in cancer immunotherapy. The study design is innovative and the findings are very interesting. The manuscript can be considered for publication after minor improvements:
- Several reports discussed the advantages of non-viral gene delivery vectors (e.g. nanoparticles) compared to their viral counterparts, especially in terms of biosafety and scalability. However, the authors adopted a viral vector in this study. The authors should refer to this argument and justify the potential benefits of their vector in comparison with nanoparticles. The authors should also discuss the possible scale-up issues if their vector will be clinically-translated.
